# Novel Genotype of HA Clade 2.3.4.4b H5N8 Subtype High Pathogenicity Avian Influenza Virus Emerged at a Wintering Site of Migratory Birds in Japan, 2021/22 Winter

**DOI:** 10.3390/pathogens13050380

**Published:** 2024-05-02

**Authors:** Berihun Dires Mihiretu, Tatsufumi Usui, Masahiro Kiyama, Kosuke Soda, Tsuyoshi Yamaguchi

**Affiliations:** 1Joint Graduate School of Veterinary Sciences, Tottori University, 4-101 Koyama-Minami, Tottori 680-8553, Japan; d22a5004x@edu.tottori-u.ac.jp (B.D.M.); soda@tottori-u.ac.jp (K.S.); yamaguti@tottori-u.ac.jp (T.Y.); 2Avian Zoonosis Research Center, Faculty of Agriculture, Tottori University, 4-101 Koyama-Minami, Tottori 680-8553, Japan; 3Natural Green Resources Division, Department of the Environment and Consumer Affairs, Tottori Prefecture, 1-220 Higashi-machi, Tottori 680-8570, Japan

**Keywords:** high pathogenicity avian influenza virus, low pathogenicity avian influenza virus, wintering site, reassortment, Japan

## Abstract

Surveillance of avian influenza virus (AIV) was conducted in the 2021–2022 winter season at a wintering site of migratory *Anatidae* in Japan. An H5N8 subtype high pathogenicity AIV (HPAIV) with a unique gene constellation and four low pathogenicity AIVs (LPAIVs) were isolated from environmental samples. The genetic origin of the HPAIV (NK1201) was determined with whole-genome sequencing and phylogenetic analyses. Six of NK1201’s eight genes were closely related to HA clade 2.3.4.4b H5N8 subtype HPAIVs, belonging to the G2a group, which was responsible for outbreaks in poultry farms in November 2021 in Japan. However, the remaining two genes, PB1 and NP, most closely matched those of the LPAIVs H7N7 and H1N8, which were isolated at the same place in the same 2021–2022 winter. No virus of the NK1201 genotype had been detected prior to the 2021–2022 winter, indicating that it emerged via genetic reassortment among HPAIV and LPAIVs, which were prevalent at the same wintering site. In addition, experimental infection in chickens indicated that NK1201 had slightly different infectivity compared to the reported infectivity of the representative G2a group H5N8 HPAIV, suggesting that the PB1 and NP genes derived from LPAIVs might have affected the pathogenicity of the virus in chickens. Our results directly demonstrate the emergence of a novel genotype of H5N8 HPAIV through gene reassortment at a wintering site. Analyses of AIVs at wintering sites can help to identify the emergence of novel HPAIVs, which pose risks to poultry, livestock, and humans.

## 1. Introduction

Avian influenza, also known as bird flu, is caused by an influenza A virus (IAV) (*Alphainfluenzavirus influenzae*) of the Orthomyxoviridae family [1]. IAVs have eight segmented genomes (PB2, PB1, PA, HA, NP, NA, M, and NS). Based on the severity of symptoms they cause in chickens, avian influenza viruses (AIVs) are classified as high and low pathogenicity AIVs (HPAIVs and LPAIVs, respectively). LPAIVs cause no clinical symptoms or mild symptoms such as ruffled feathers or a drop in egg production, whereas HPAIVs cause severe disease and high mortality. IAVs are further classified into subtypes based on two of their surface proteins, hemagglutinin (HA) and neuraminidase (NA) [2]. So far, 16 HA and 9 NA subtypes have been detected in birds [3]. Hence, the combination of any of these subtypes can happen, leading to the generation of new reassorted viruses [4]. The evolution of novel IAVs may exacerbate their impact on public health and the poultry industry. For instance, reassorted IAVs containing genes derived from avian-origin viruses were responsible for four previously reported influenza pandemics (1918 H1N1, 1957 H2N2, 1968 H3N2, and 2009 H1N1) [5,6,7]. In addition, the novel H5N1, H5N2 and H5N8 subtype HPAIVs generated by reassortment among Asian H5N8 HPAIV and North American LPAIVs resulted in the culling of millions of chickens and turkeys in the USA in 2014 [8]. The transmission of avian IAV subtypes to mammals, such as pigs [9] and minks [10], which may facilitate the evolution of avian IAV, has been reported. Avian IAVs can also cause outbreaks and high mortality in domestic poultry [11].

Migratory birds are the natural reservoir of IAVs. Because of their movements, they play a pivotal role in the global spread and reassortment of IAVs [11,12,13]. When two different influenza viruses co-infect a host cell, there is a possibility of genetic exchange (reassortment) during virus assembly due to the segmented nature of influenza A viral genomes. Several such reassortments among HPAIVs and LPAIVs have been reported. Between 2011 and 2016, four reassorted viruses containing two to five LPAIV genes (H5N2 clade 2.3.4, H5N8 clade 2.3.4.4b, H5N6 clade 2.3.4.4 group C, and H5N8 clade 2.3.4.4 group B HPAIVs) were isolated in China, South Korea, Vietnam, and Russia, respectively [14,15,16,17]. However, it is unclear when and where all of these reassortment events occurred, because the putative parent HPAIVs and LPAIVs were isolated from different locations and in different seasons. These observations suggest that the majority of genetic reassortments probably occur among migratory birds returning to their nesting sites from wintering sites around the world.

In this study, we surveyed environmental water and fecal samples for AIVs at a wintering site of migratory waterfowls in Tottori, Japan, during the winter and spring migration [18] of one season (December 2021 to March 2022) and isolated HPAIV A/water/Tottori/NK1201-2/2021 (H5N8) (NK1201) [19]. Recently, various genotypes of H5Nx clade 2.3.4.4b HPAIVs [19,20,21,22,23,24] and H7N7 LPAIVs [24] have been detected in Japan. H5N8 and H5N1 HPAIVs isolated in the 20221–2022 winter season in Japan were divided into three groups (G2a, G2b, and G2d) based on their HA gene sequences [19]. Moreover, representative isolates from the G2a, G2b, and G2d groups were found to have different infectivities in chickens [21]. The H5N8 viruses belonged to the G2a group and caused HPAI in two poultry farms in Akita and Kagoshima prefectures in November 2021 [19,25]. The origin of the NK1201 HPAIV isolated on 1 December in our surveillance is unknown. It was found to be an H5N8 subtype G2a group virus, but had PB1 and NP genes derived from LPAIVs. To characterize the genetic origin of NK1201, phylogenetic analyses were conducted for all eight gene segments of NK1201 and the LPAIVs isolated at the same site and same season. Then, the infectivity of NK1201 in chickens was examined to clarify the effects of gene exchange.

## 2. Materials and Methods

### 2.1. Surveillance of Avian Influenza Virus Using Environmental Samples

A total of 14 environmental water samples (1 L each) from the surface of the reservoirs and 140 fecal samples were collected from a wintering site of migratory birds (ducks, geese and swans) at Nikko, Tottori prefecture, Japan (35.514125, 134.066304), during the period 1 December 2021 to 23 March 2022 (Figure 1 and Table 1). The water bottles were placed on frozen ice packs immediately after collection and maintained in the cold chain during transportation to the laboratory. The water samples were processed immediately for virus isolation upon arrival. To concentrate the influenza virus in the water samples, we adsorbed the virus with chicken red blood cells (cRBCs) as described previously [26], with some modifications. Briefly, 1 L of each water sample was transferred into a sterile 1 L plastic bottle (Sanplatec, Osaka, Japan) through a 106 μm mesh filter to remove debris. Ten milliliters of 10% formalin-fixed cRBCs was added to the water samples and mixed by inverting the bottles. The water samples were incubated for 1 h at 4 °C, transferred to 250 mL NALGENE PPCO centrifuge bottles (Thermo Fisher Scientific, Waltham, MA, USA), and centrifuged at 3000× *g* at 4 °C for 5 min. The precipitated cRBCs were resuspended with 2 mL of chilled phosphate-buffered saline (PBS), pH 7.4, containing 10,000 units/mL penicillin (Meiji Seika Pharma, Tokyo, Japan) and 10,000 μg/mL streptomycin (Meiji Seika Pharma) (PBS-PS) that were prepared in the laboratory. The entire volume (up to 8 mL) was inoculated into the allantoic cavities of four 9- to 11-day-old embryonated chicken eggs (ECEs) (Aoki Breeder Farm, Tochigi, Japan) with equal volumes of the mixture. In order to isolate viruses from fecal samples, feces were suspended in PBS-PS. The suspended fecal samples were centrifuged at 1600× *g* at 4 °C for 5 min, followed by 20,000× *g* at 4 °C for 5 min. A 0.2 mL aliquot of each fecal sample supernatant was inoculated into the allantoic cavities of two 9- to 11-day-old ECEs. The ECEs inoculated with water and fecal samples were incubated at 37 °C for 48 h or until they died, and were then chilled at 4 °C overnight. The allantoic fluid from each egg was collected separately and the presence of hemagglutinating virus was tested with a hemagglutination (HA) test. All HA-positive allantoic fluids were tested with a rapid diagnosis kit for IAV (ESPLINE INFLUENZA A&B-N, Fujirebio, Tokyo, Japan). When viruses were not recovered, the collected allantoic fluid was inoculated into ECEs for passage. The HA and NA subtypes of the isolates were determined with a hemagglutinin inhibition (HI) test using 16 HA reference antisera and reverse transcriptase polymerase chain reaction (RT-PCR) using specific primers for each of the nine NA subtypes of IAVs.

### 2.2. Next Generation Sequencing for HPAIV

To obtain the complete genome of a water-derived HPAIV A/water/Tottori/NK1201-2/2021 (H5N8), RNA was extracted using the Quick-RNA viral kit (Zymo Research, Irvine, CA, USA) according to the manufacturer’s instructions. RNA samples were sent to Bioengineering Lab. Co., Ltd. (Kanagawa, Japan), processed using the MGIEasy RNA Directional Library Prep set (MGI Tech Co., Ltd., Guangdong, China) and sequenced by the paired-end method with DNBSEQ-G400 using the DNBSEQ-G400RS High-throughput Sequencing Kit (MGI Tech Co., Ltd.). Consensus sequences were generated by using the Automatic Influenza virus Genome Assembly and subtyping System (FluGAS) software version 2.1 (World Fusion, Tokyo, Japan).

### 2.3. RNA Extraction and PCR Amplification of LPAIVs

Viral RNAs were extracted from IAV-positive allantoic fluids using a QIAamp Viral RNA Mini kit (Qiagen, Hilden, Germany), in accordance with the manufacturer’s instructions. A Uni-12 primer (5′-AGCAAAAGCAGG-3′) was used to reverse transcribe viral RNA to complementary DNA (cDNA), as described by Hoffmann et al. [27]. All eight genes of the isolate were amplified by PCR using GoTaq Green master mix DNA polymerase (Promega, Madison, WI, USA) and segment-specific primers. Amplified genes were separated by gel electrophoresis, the gel bands were cut out and DNA was extracted from the bands with a MinElute Gel Extraction kit (Qiagen). Then, each segment was directly sequenced using a Genetic analyzer 3130 (Applied Biosystems, Foster City, CA, USA). PCR products were used as a template for sequencing with a BigDye terminator cycle-sequencing kit version 3.1 (Applied Biosystems). After sequencing, nucleotide editing and sequence assembly were conducted using Codon-Code Aligner version 10.0.1 (CodonCode, Dedham, MA, USA). All nucleotide sequences were deposited in the NCBI GenBank/EMBL/DDBJ and can be accessed using their corresponding accession numbers (Table 1).

### 2.4. Phylogenetic Analysis

Phylogenetic trees were constructed using the maximum likelihood method with the general time-reversible model and 1000 bootstrap replicates in Molecular Evolutionary Genetics Analysis (MEGA) version X [28]. For phylogenetic analysis, we used the open reading frames of the 8 gene segments of the AIVs isolated in the current study and nucleotide sequences from public databases as of October 2023.

### 2.5. Species Identification from Fecal Samples

Genomic DNA was extracted from fecal samples positive for IAVs using a QIAamp DNA Stool mini kit (Qiagen). Then, the mitochondrial cytochrome c oxidase subunit I (COI) gene was amplified and sequenced using the Genetic analyzer 3130. The nucleotide sequences of COI were used for species identification with the Barcode of Life Data System version 4 [29].

### 2.6. Pathogenicity Test of H5N8 HPAIV in Chickens

The pathogenicity of NK1201 to chickens was examined via intravenous inoculation in a biosafety level 3 facility at the Avian Zoonosis Research Center, Tottori University, Japan. Eight six-week-old chickens negative for the H5-specific antibody according to the HI test were subjected to wing vein intravenous inoculation with 0.2 mL of infectious allantoic fluid diluted ten-fold with sterile PBS. The clinical signs of chickens were observed daily to determine the mortality within 10 days, in accordance with the WOAH manual (https://www.woah.org/fileadmin/Home/eng/Health_standards/tahm/3.03.04_AI.pdf. accessed on 21 April 2023). The dose of inoculum was 6 × 10^6^ fifty-percent egg infectious dose (EID_50_).

Next, to determine the 50% chicken lethal dose (CLD_50_), 16 six-week-old female chickens were divided into 4 groups (4 in each group) and placed in an isolator. Feed and water were provided ad libitum. The chickens in group 1, 2, 3 and 4 were intranasally inoculated with 10^3^, 10^4^, 10^5^ and 10^6^ EID_50_ of NK1201, respectively. Oropharyngeal and cloacal swabs were collected at 1, 3, 5, 7 and 10 days post inoculation (dpi) or when the chicken died. Virus isolation and the HA test were conducted according to the method mentioned above. 

## 3. Results

### 3.1. Isolation of HPAIV and LPAIV from Water Samples

Of the 14 water samples collected at the wild bird habitats at Nikko during the 2021–2022 winter season, two tested positive for AIVs. The viruses were identified as H5N8 and H3N8 subtypes and designated as NK1201 and NK1214 (Table 1). A multibasic amino acid sequence (PLREKRRKR/GLF) was found at the HA cleavage site of the NK1201 isolate, which shows that it was HPAIV (Table 1).

### 3.2. Isolation of LPAIV from Fecal Samples

Of the 140 fecal samples collected at Nikko, three were confirmed to be positive for AIVs. The H7N7, H1N8, and H6N2 subtype viruses were designated as NK12F18, NK2F2, and NK3F4 (Table 1). Based on the nucleotide sequences of mitochondrial COI, NK12F18 and NK3F4 were isolated from *Anas* duck species and NK2F2 was isolated from an *Anser* goose species. The HA cleavage site of NK12F18 (H7N7) had a single arginine (PEIPKGR/GLF), suggesting that it was LPAIV (Table 1).

### 3.3. Nucleotide Comparison and Phylogenetic Analysis of NK1201

The influenza virus closely related to NK1201 was cross-referenced with sequences from public databases (Table 2 and Appendix A). Six genes (PB2, PA, HA, NA, M, and NS) of the NK1201 isolated in the present study had high similarities with the HPAIVs A/environment/Akita/TU1-49/2021 (H5N8), A/chicken/Akita/7C/2021 (H5N8), A/chicken/Kagoshima/B3T/2021 (H5N8) and A/chicken/Kagoshima/TU3-46.47/2021 (H5N8), which were responsible for outbreaks in two poultry farms in Japan in November 2021 [19,25]. Two internal genes, PB1 and NP, of NK1201 had high nucleotide identity with the PB1 and NP LPAIV genes of A/eurasian coot/Shandong/W22/2022 (H8N4) and A/Eurasian wigeon/Shanghai/NH112330/2021 (H9N2) obtained from public databases, respectively, and had a higher nucleotide and amino acid identity with genes of the LPAIVs isolated in this study. The PB1 gene of NK1201 had 99.60% nucleotide and amino acid identity with the NK2F2 PB1 gene, and 98.99% nucleotide and 99.33% amino acid identity with the NK12F18 PB1 gene. The NP gene of NK1201 had 99.47% nucleotide and 99.79% amino acid identity with the NP gene of NK12F18 (Table 2). NK12F18 was isolated from an *Anas* species fecal sample collected on the same day as NK1201 (Table 1). NK2F2 was isolated from an *Anser* species fecal sample collected from the same location 2 months later. Unexpectedly, all eight genes of NK1201 showed high nucleotide and amino acid identity with A/Whooper swan/Korea/21WC116/2022 (H5N8), which was isolated less than two months after the isolation of NK1201.

Phylogenetic analysis showed that the HA gene of NK1201 belongs to the HA clade 2.3.4.4b G2a group (Figure 2a). PB1 of the HPAIV NK1201 clustered with the PB1 genes of LPAIV NK12F18, NK2F2, A/eurasian coot/Shandong/W22/2022 (H8N4), and A/Whooper swan/Korea/21WC116/2022 (H5N8) (Figure 2b). The NP gene of HPAIV NK1201 clustered with the NP genes of the LPAIV NK12F18, A/Eurasian wigeon/Shanghai/NH112330/2021 (H9N2), and A/Whooper swan/Korea/21WC116/2022 (H5N8) (Figure 2c), whereas the remaining six genes clustered with the HPAIVs A/environment/Akita/TU1-49/2021 (H5N8), A/chicken/Kagoshima/TU3-46.47/2021 (H5N8) and A/chicken/Kagoshima/B3T/2021 (H5N8) (Figure 2a and Appendix A).

Additionally, the phylogenetic analysis showed that gene segments were shared among the different subtype LPAIVs isolated between December 2021 and March 2022 at the Nikko wintering site. As well as PB1 genes, the PB2 genes of the NK12F18 (H7N7) and NK2F2 (H1N8) LPAIVs clustered together. The M and PA genes of the NK1214 (H3N8) LPAIV clustered with those of the NK12F18 and NK2F2 LPAIVs, respectively (Figure 2b, Appendix A and Figure 3).

### 3.4. Pathogenicity of NK1201 in Chickens

After intravenous inoculation with the NK1201 isolate, all eight chickens died within 2 dpi, confirming its high pathogenicity. All chickens intranasally inoculated with 10^3^ EID_50_ and three out of the four chickens inoculated with 10^4^ EID_50_ survived throughout the experiment period. All chickens inoculated with 10^6^ and 10^5^ EID_50_ died within 4 and 7 days, respectively (Figure 4a). Consequently, CLD_50_ of NK1201 was determined to be 10^4.33^ EID_50_. Virus shedding was examined in oropharyngeal and cloacal swabs of 10^4^ and 10^5^ EID_50_ inoculated chickens. All chickens that survived the experimental period did not shed the virus. The highest virus shedding was observed at 5 dpi in both oropharyngeal and cloacal swabs (Figure 4b,c).

NK1201 was a unique reassorted H5N8 HPAIV isolated from water samples at the wintering site of migratory birds in Japan during the 2021–2022 winter season. Whole-genome sequencing and phylogenetic analysis showed that NK1201 had two genes (PB1 and NP) derived from LPAIVs isolated from waterfowl fecal samples at the same site and season. The high pathogenicity of NK1201 was confirmed through experimental infection in chickens, and the CLD_50_ was determined to be 10^4.33^ EID_50_.

## 4. Discussion

Although reassortment is essential for the evolution of IAVs, it is rarely known when and where a reassortant virus first appeared. Most of the events may happen at the nesting site of migratory birds since many of the genes of reassortant viruses are ancestral to viruses with different isolation times and locations. In this study, we found evidence of reassortment events in H5N8 HPAIV at a wintering site in the 2021/22 winter. Six of the eight genes of NK1201 were closely related to those of H5N8 HPAIVs that caused outbreaks in poultry farms in Japan in November 2021. These viruses were of the E2 genotype H5N8 virus lineage, which were prevalent in the previous 2020/21 winter season in Japan, South Korea, China and Europe [19], but not detected in the 2021 summer in Japan. Based on the public database, the PB1 and NP genes of NK1201 showed high similarity to those of Chinese LPAIVs, A/eurasian coot/Shandong/W22/2022 (H8N4) and A/Eurasian wigeon/Shanghai/NH112330/2021 (H9N2), which were isolated in 2021/22 winter; this implies that the LPAIVs, along with HPAIVs, might have been brought to Japan by an autumn migration. Interestingly, phylogenetic analysis revealed that the PB1 and NP genes of NK1201 were closely related to LPAIVs isolated from waterfowl fecal samples collected in the same season and from the same location. The genotype of NK1201 was unique and had not been detected during that winter season in Japan, suggesting that NK1201 emerged via genetic reassortment at this sampling site in the 2021 winter. Moreover, phylogenetic analysis also showed that A/Whooper swan/Korea/21WC116/22 (H5N8), with a gene constellation identical to that of NK1201, was detected in South Korea less than two months after the isolation of NK1201. This demonstrates that the newly emerged NK1201 at the Nikko wintering site in Japan was immediately disseminated to South Korea by migratory birds.

The HA gene of NK1201 clustered with G2a group H5N8 HPAIVs, which were responsible for an outbreak in layer chickens in Akita (northern Japan) and Kagoshima (southern Japan), Japan, at the beginning of the 2021 winter season [19,25]. The lethality of the virus to chickens intranasally inoculated with 10^6^ EID_50_/0.1 mL of NK1201 was comparable to that of the representative isolate A/chicken/Akita/7C/2021 (H5N8) used to evaluate the infectivity of G2a groups [21]. However, NK1201 had slightly lower infectivity with CLD_50_ (10^4.33^ EID_50_) in chickens compared to A/chicken/Akita/7C/2021 (H5N8) (10^3.83^ EID_50_) [21]. Since chickens of different ages were used in the present study, the slight difference in CLD_50_ might be due to the differences among the experimental chickens. The two internal genes (PB1 and NP) of NK1201 derived from LPAIVs might have also affected the infectivity and proliferation of the virus in chickens. All eight genes of A/environment/Akita/TU1-49/2021 (H5N8) (AkitaTU1-49), a virus isolated from the same farm where A/chicken/Akita/7C/2021 (H5N8) caused an outbreak, are available in the database. Thus, the PB1 and NP amino acids of AkitaTU1-49 were compared to those of NK1201. Amino acid differences were observed at various positions in the PB1 and NP genes of NK1201 and AkitaTU1-49. Among these differences, methionine (M) was found at position 105 of NP in NK1201 and valine (V) was found at the same position in AkitaTU1-49. Tada et al. [30] found that the polymerase activity of an H5N1 HPAIV with V105 in chicken cells was greater than that of another H5N1 HPAIV with M105, so V105 might be one of the determinants for the adaptation of AIVs from ducks to chickens. Thus, the M at position 105 in the NP protein may lead to a decrease in the infectivity of NK1201 in chickens.

The Nikko wintering site is a marshy rice field with shallow ponds and is visited by a large number of migratory birds, including various species of ducks, geese and swans, for wintering. In previous years, several HPAIVs and LPAIVs have also been isolated from fecal samples and environmental water at this wintering site. In the present study, reassortment occurred not only among HPAIV and LPAIVs, but also within different subtype LPAIVs. Reassortment was observed among the LPAIV PB2 and PB1 genes of NK2F2 and NK12F18, the PA gene of NK2F2 and NK1214, and the M gene of NK12F18 and NK1214 (Figure 2b and Appendix A). With the exception of N8 NA, only NS genes of the five Nikko isolates were derived from different origins (Appendix A). The phylogenetic trees also show a variety of LPAIV lineages that were introduced to the Nikko site from South Korea, Russia, China, and North America (Figure 2b,c and Appendix A), demonstrating that AIV gene pools at the Nikko wintering site contributed to the evolution of novel reassortant viruses.

In the 2020/21 winter, a reassortant H5N8 HPAIV containing the PA gene derived from LPAIVs found at the same location was reported to emerge at another wintering site in Japan [22]. Our findings and those of Okuya et al. [24] indicate that novel gene reassortment events of AIVs can emerge not only in northern nesting areas of migratory water birds, but also at overwintering sites of migratory birds. Every winter, influenza virus is brought to the island nation of Japan along with the migration of birds. Satellite tracking technology revealed a new flyway for migratory birds in East Asia [31], with the geese from the *Izumo* plain (Shimane prefecture) in southwestern Japan directly crossing over the Sea of Japan and arriving at the east coast of North Korea [32]. Tundra swan from *Nakaumi* Lagoon (Tottori prefecture) also cross over the Sea of Japan [33]. With the migration of migratory birds, the influenza virus can cross the Sea of Japan in just one night. Overlapping flyways of migration on the eastern coast of Eurasia allow diverse influenza viruses to come and go each winter in East Asia. To monitor the spread of influenza viruses by migration across continents, Japan is a particularly important area in East Asia due to its island characteristics, where the virus is basically brought in only by various migratory birds. This study suggests that a novel reassortant HPAIV emerged from LPAIVs and HPAIVs introduced from the Eurasia continent across the Sea of Japan and immediately spread to other countries.

## 5. Conclusions

In this study, we discovered that a novel reassortant HPAIV emerged at the Nikko wintering site in Japan. The newly emerged H5N8 HPAIV had a unique gene constellation and may have lower infectivity in chickens than other closely related viruses, as inferred from the experimental infection results and reported genetic characteristics. This shows that wintering sites for migratory birds can play a critical role in the evolution and spread of novel reassortant viruses, which may cause public health problems in addition to economic losses in poultry production. Our findings stress that the continuous analysis of LPAIV isolates and HPAIVs at wintering sites is important in recognizing influenza gene pool spots that contribute to the emergence of novel HPAIVs.

## Figures and Tables

**Figure 1 pathogens-13-00380-f001:**
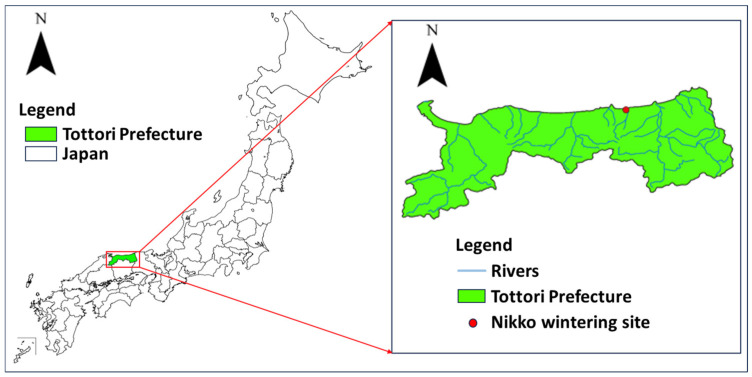
Location of Nikko wintering site of migratory birds in Tottori Prefecture, Japan.

**Figure 2 pathogens-13-00380-f002:**
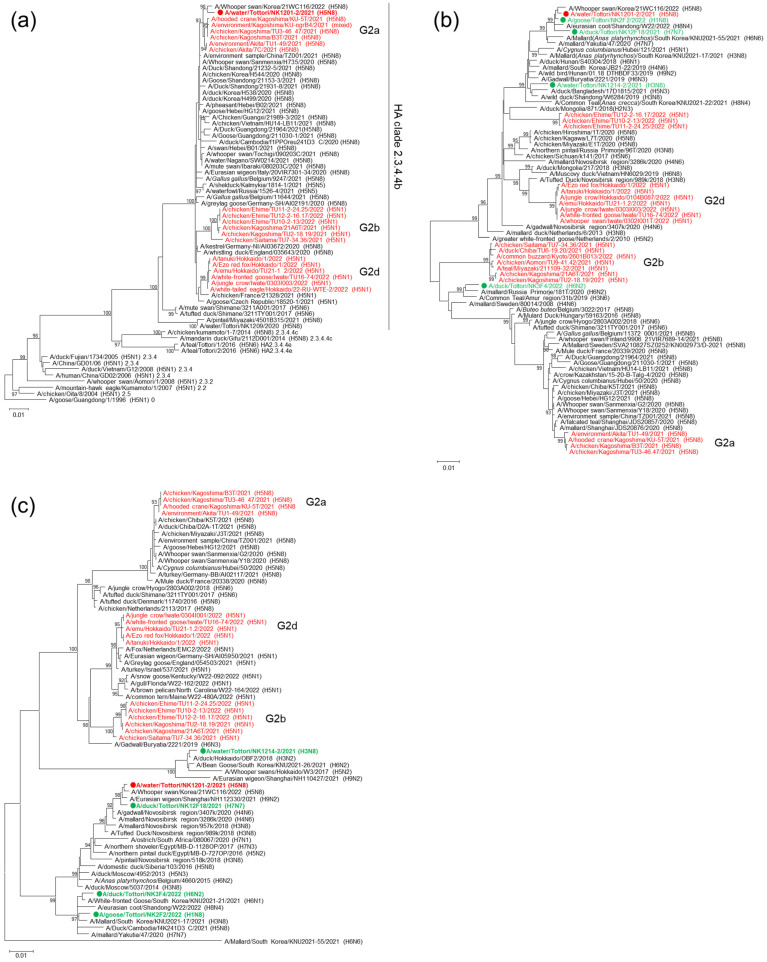
Phylogenetic trees based on ORF of the HA, PB1 and NP gene segments of the NK1201 HPAIV and LPAIVs isolated at the Nikko wintering site. HA (**a**), PB1 (**b**) and NP (**c**) trees were constructed using the maximum likelihood method with the general time-reversible model and 1000 bootstrap replicates. Bootstrap values > 90% are shown at the nodes. Red: H5N8 HPAIVs isolated during 2021–2022 in Japan; Green: LPAIV isolates from wintering site for migratory birds in Tottori Prefecture. Filled circle indicates isolates in this study.

**Figure 3 pathogens-13-00380-f003:**
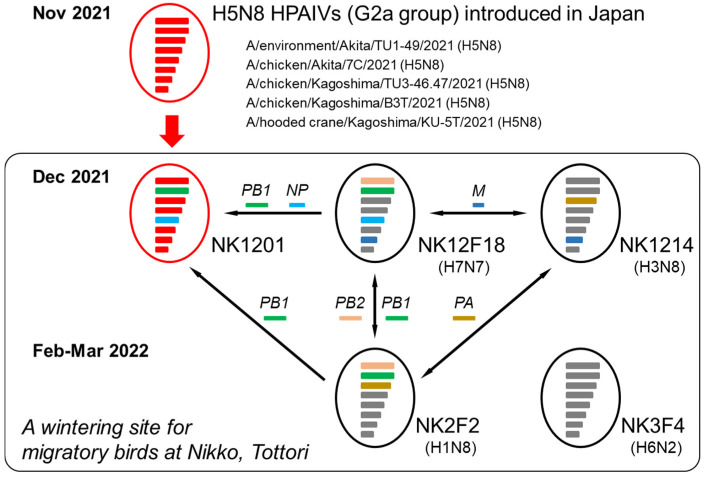
Schematic presentation of HPAIV and LPAIV gene reassortment at wintering site of migratory birds in Tottori Prefecture, Japan. The PB1 gene of NK1201 derived from LPAIV and closely related to NK2F2 and NK12F18 is labeled in green. The NP gene of NK1201 derived from LPAIV and closely related to NK12F18 is labelled in blue. All six genes of NK1201 were derived from HPAIV and closely related to the H5N8 (G2a) HPAIVs introduced in Japan in 2021/2022 season are labelled in red. The PA gene shared by NK2F2 and NK1214 and M gene shared by NK12F18 and NK1214 are labelled in dark yellow and dark blue, respectively.

**Figure 4 pathogens-13-00380-f004:**
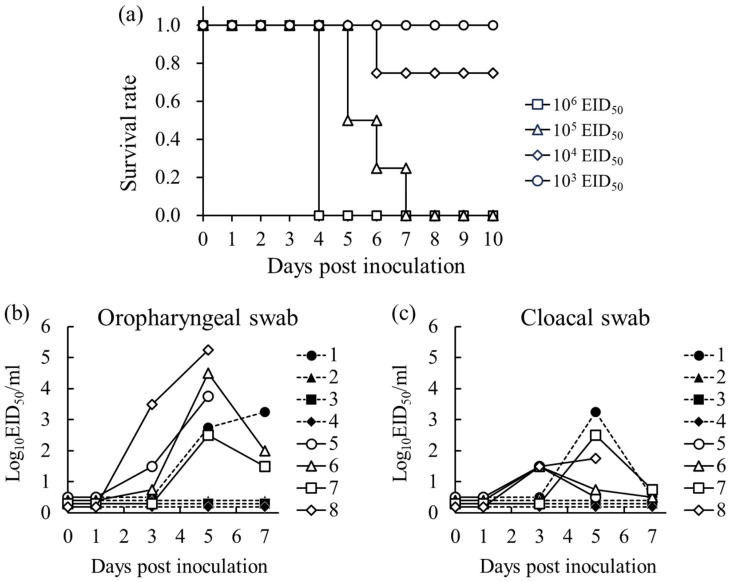
Survival rate of chickens intranasally inoculated with various infectious doses (10^3^ to 10^6^ EID_50_) of NK1201 (**a**). Virus shedding in oropharyngeal swabs (**b**) and cloacal swabs (**c**) of chickens intranasally inoculated with 10^4^ EID_50_ (no. 1–4) and 10^5^ EID_50_ (no. 5–8) of NK1201.

**Table 1 pathogens-13-00380-t001:** HPAIV and LPAIV isolates identified in a survey of influenza A at Nikko, Tottori Prefecture, Japan.

Collection Date	Samples		Isolates	Abbreviations	Putative Sequence of HA Cleavage Site	Remark	Accession Number
	Water	Feces					
2021-12-01	2	20	A/water/Tottori/NK1201-2/ 2021 (H5N8)	NK1201	PLREKRRKR/GLF	HPAIV	LC699141–LC699148
			A/duck/Tottori/NK12F18/ 2021 (H7N7)	NK12F18	PEIPKGR/GLF	LPAIV	LC719253–LC719260
2021-12-14	2	20	A/water/Tottori/NK1214-2/ 2021 (H3N8)	NK1214	PEKQTR/GLF	LPAIV	LC721722–LC721729
2022-01-06	2	20	Negative for AIV	n/a	n/a	n/a	n/a
2022-01-26	2	20	Negative for AIV	n/a	n/a	n/a	n/a
2022-02-10	2	20	A/goose/Tottori/NK2F2/2022 (H1N8)	NK2F2	PSIQSR/GLF	LPAIV	LC714903–LC714910
2022-03-03	2	20	A/duck/Tottori/NK3F4/2022 (H6N2)	NK3F4	PQIENR/GLF	LPAIV	LC721730–LC721737
2022-03-23	2	20	Negative for AIV	n/a	n/a	n/a	n/a
Total	14	140	5				

n/a: All samples were negative for AIVs.

**Table 2 pathogens-13-00380-t002:** Nucleotide and amino acid similarities of the NK1201 two internal genes to most closely related influenza viruses.

Genes	Closely Related Viruses	Collection Date	Nucleotide Identity (%)	Amino Acid Identity (%)
Polymerase basic 1 (PB1)	A/goose/Tottori/NK2F2/2022 (H1N8) *	2022-2-10	99.60 (2265/2274)	99.60
	A/Whooper swan/Korea/21WC116/2022 (H5N8)	2022-1-23	99.47 (2262/2274)	99.47
	A/eurasian coot/Shandong/W22/2022 (H8N4)	2022-1-16	99.30 (2258/2274)	99.60
	A/mallard/Yakutia/47/2020 (H7N7)	2020-8-13	99.03 (2252/2274)	99.47
	A/duck/Tottori/NK12F18/2021 (H7N7) *	2021-12-01	98.99 (2251/2274)	99.33
	A/Mallard(*Anas platyrhynchos*)/South Korea/KNU2021-55/2021 (H6N6)	2021-9-09	98.59 (2242/2274)	99.33
	A/wild bird/Hunan/01.18 DTHBDF33/2019 (H9N2)	2019-1-18	98.28 (2235/2274)	99.07
	A/duck/Hunan/S40304/2018 (H6N1)	2018-11-13	98.20 (2233/2274)	99.60
	A/mallard/South Korea/JB21-22/2019 (H4N6)	2019-9-28	98.11 (2231/2274)	99.47
	A/Gadwall/Buryatia/2221/2019 (H6N3)	2019-10-12	98.11 (2231/2274)	99.47
Nucleocapsid protein (NP)	A/Whooper swan/Korea/21WC116/2022 (H5N8)	2022-1-23	99.67 (1492/1497)	99.79
	A/Eurasian wigeon/Shanghai/NH112330/2021 (H9N2)	2021-11-23	99.59 (1454/1460)	99.58
	A/duck/Tottori/NK12F18/2021 (H7N7) *	2021-12-01	99.47 (1489/1497)	99.79
	A/gadwall/Novosibirsk region/3407k/2020 (H4N6)	2020-8-29	99.33 (1487/1497)	99.79
	A/mallard/Novosibirsk region/3286k/2020 (H4N6)	2020-8-29	99.27 (1486/1497)	99.79
	A/Tufted Duck/Novosibirsk region/989k/2018 (H3N8)	2018-10-01	98.93 (1481/1497)	99.59
	A/mallard/Novosibirsk region/957k/2018 (H3N8)	2018-9-28	98.73 (1478/1497)	99.59
	A/Anas platyrhynchos/Belgium/4660/2015 (H6N2)	2015-9-18	97.73 (1463/1497)	98.99
	A/pintail/Novosibirsk region/518k/2018 (H3N8)	2018-10-01	97.73 (1463/1497)	99.79
	A/Mallard/Netherlands/14017570/2014 (H11N9)	2014-12-17	97.66 (1462/1497)	99.39

*: LPAIVs isolated in this study.

## Data Availability

The generated sequence data were deposited in the NCBI GenBank/EMBL/DDBJ database.

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
