# Peer review of "Novel Genotype of HA Clade 2.3.4.4b H5N8 Subtype High Pathogenicity Avian Influenza Virus Emerged at a Wintering Site of Migratory Birds in Japan, 2021/22 Winter"

_pathogens, 2024, doi:10.3390/pathogens13050380_

Round 1

Reviewer 1 Report

Comments and Suggestions for Authors

The key point in the manuscript, that the pathogenicity about the NK1201 is lower that the H5N8 HPAIV in the same genotype, is not acceptable for me. The authors opinion was from the CLD50 of two viruses, the CLD50 of NK1201 was 104.33 EID50, while the CLD50 of the reference isolate was 103.83 EID50. Firstly, the experimental infection in chickens were not carried out at the same time, the animals were not the SPF chickens from the same batch, so the differentiation of CLD50 between two viruses might induced by the animal; Secondly, the differentiation of two CLD 50 was just 100.5EID50, which is too low to be determined as significant difference. So the putative opinion in the study that the PB1 and NP might affect the infectivity of virus in chicken is not sufficient either. 

Reviewer 2 Report

Comments and Suggestions for Authors

The present research manuscript reports a novel genotype of H5N8 avian influenza virus in wild birds in Japan. This is an important finding  and would be useful for the relevant scientific community engaged in infectious diseases research. While Results, Discussions, and Conclusions are well written, the Introduction and Methods sections need more information. I have following suggestions for the authors to consider:

1. In the paragraph after Line 50 in the Introduction section, please add that various avian IAV subtypes have been reported to be transmitted to mammalian species, such as pigs (DOI: 10.1007/s11262-022-01904-w) and minks (DOI: 10.1080/22221751.2021.1899058 ), facilitating the evolution of IAV. Also, avian IAV subtypes may be transmitted to the domestic poultry (such as Gallus gallus domesticus), threatening the disease outbreaks and inflicting severe mortality in chicken flocks (https://doi.org/10.3390/pathogens10050630). 

2.  Lines 66 and 67: Why did the surveillance take place in December and March months? Is this the time of the year when waterfowls migrate from or to Japan? Please explain briefly, with citations.

3. In Introduction section, please add more information on the circulation of novel genotypes of avian IAV in Japan (not necessarily H5N8), using recently published literature, to justify this study.

4.  Methods: Line 81: Were the water samples collected from the surface of the reservoirs? Please provide details. Were the water bottles placed on frozen ice packs immediately after collection and maintained the cold chain during transportation to the laboratory? Please provide details. 

5. Methods: Line 87: Why 106 μm mesh filter?

6. Were the water samples processed immediately for virus isolation upon arrival to the laboratory or incubated for a certain period?

7. Line 97: PBS-PS? Please provide details if it was made in the laboratory or purchased in solution (pH ?)? Please also provide citations if previously reported methods were followed.

8. Methods: Paragraph 111 - 120: Please mention clearly if Illumina sequencing was done?

9. Methods: Paragraph 122 - 137: Please provide the details in a supplementary file about the PCR programs used, number of bands visualized on agarose gel for each gene segment of avian IAV. Was there any non-specific amplifications issues etc.? Please also provide the details of PCR reaction mixture set up - final volume used comprising dNTPs, MgCl2 concentrations etc. 

10. Please provide a brief concluding paragraph in Results section before moving on to the Discussion. 

Reviewer 3 Report

Comments and Suggestions for Authors

Review report:

In this study, authors surveyed environmental water and fecal samples for AIVs at a wintering site of migratory waterfowls in Tottori, Japan. One H5N8 subtype high pathogenicity AIV (HPAIV) with a unique gene constellation and four low pathogenicity AIV (LPAIVs) were isolated. To further characterize the genetic origin of NK1201, phylogenetic analyses were conducted for all eight gene segments of NK1201 and LPAIVs isolated at the same site and same season. Then, the infectivity of NK1201 in chicken was examined to clarify the effects of gene exchange. These studies are helpful for understanding the evolution and spread of novel reassortant viruses.

Altogether this data is interesting, well organized the figures and well written the manuscript.

Minor comment:

1.Please write clearly figure legends of supplementary data.

Author Response

Revision notes.

Reviewer 3 questions and answers

Thank you for your thorough review of our manuscript. We sincerely appreciate the time and effort you invested in providing constructive feedback.

  1. Please write clearly figure legends of supplementary data.

Answer.  

Full name of the genes in the supplementary file are included (Line 353-354).

The explanation of the G2a, G2b, and G2d classifications based on HA and the length of the coding part for each of the five segments are added to the supplementary file legend.

The following references are included to the revised manuscript.

Chauhan, R. P., & Gordon, M. L. (2022). An overview of influenza A virus genes, protein functions, and replication cycle highlighting important updates. Virus Genes58(4), 255-269.

Sun, H., Li, F., Liu, Q., Du, J., Liu, L., Sun, H., ... & Liu, J. (2021). Mink is a highly susceptible host species to circulating human and avian influenza viruses. Emerging microbes & infections10 (1), 472-480.

Onuma Manabu, Kakogawa Masayoshi, Yanagisawa Masae, Haga Atsushi, Okano Tomomi, Neagari Yasuko, Okano Tsukasa, Goka Koichi, and Asakawa Mitsuhiko (2017). Characterizing the temporal patterns of avian influenza virus introduction into Japan by migratory birds. Journal of Veterinary Medical Science 79, no. 5: 943-951.

Okuya Kosuke, Esaki Mana, Tokorozaki Kaori, Hasegawa Taichi, and Ozawa Makoto (2024). Isolation and genetic characterization of multiple genotypes of both H5 and H7 avian influenza viruses from environmental water in the Izumi plain, Kagoshima prefecture, Japan during the 2021/22 winter season. Comparative Immunology, Microbiology and Infectious Diseases 109: 102182.

Isoda Norikazu, Onuma Manabu, Hiono Takahiro, Ivan Sobolev, Hew Yik Lim, Nabeshima Kei, Honjyo Hisako, Yokoyama Misako, Alexander Shestopalov, and Sakoda Yoshihiro (2022). Detection of new H5N1 high pathogenicity avian influenza viruses in winter 2021–2022 in the far east, which are genetically close to those in Europe. Viruses 14, no. 10: 2168.

Mine Junki, Takadate Yoshihiro, Kumagai Asuka, Sakuma Saki, Tsunekuni Ryota, Miyazawa Kohtaro, and Uchida Yuko (2024). Genetics of H5N1 and H5N8 High-Pathogenicity Avian Influenza Viruses Isolated in Japan in Winter 2021–2022. Viruses 16, no. 3: 358.

Round 2

Reviewer 2 Report

Comments and Suggestions for Authors

No further comments.